# Low-Cost Sensors State Estimation Algorithm for a Small Hand-Launched Solar-Powered UAV

**DOI:** 10.3390/s19214627

**Published:** 2019-10-24

**Authors:** An Guo, Zhou Zhou, Xiaoping Zhu, Fan Bai

**Affiliations:** 1School of Aeronautics, Northwestern Polytechnical University, Xi’an 710072, China; guoanuav@mail.nwpu.edu.cn (A.G.); bfaner@mail.nwpu.edu.cn (F.B.); 2Science and Technology on UAV Laboratory, Northwestern Polytechnical University, Xi’an 710072, China; zhuxp@nwpu.edu.cn

**Keywords:** low-cost sensor, state estimation, extended Kalman filter (EKF), three-stage series, full-state direct, full-state indirect, model calibration

## Abstract

In order to reduce the cost of the flight controller and improve the control accuracy of solar-powered unmanned aerial vehicle (UAV), three state estimation algorithms based on the extended Kalman filter (EKF) with different structures are proposed: Three-stage series, full-state direct and indirect state estimation algorithms. A small hand-launched solar-powered UAV without ailerons is used as the object with which to compare the algorithm structure, estimation accuracy, and platform requirements and application. The three-stage estimation algorithm has a position accuracy of 6 m and is suitable for low-cost small, low control precision UAVs. The precision of full-state direct algorithm is 3.4 m, which is suitable for platforms with low-cost and high-trajectory tracking accuracy. The precision of the full-state indirect method is similar to the direct, but it is more stable for state switching, overall parameters estimation, and can be applied to large platforms. A full-scaled electric hand-launched UAV loaded with the three-stage series algorithm was used for the field test. Results verified the feasibility of the estimation algorithm and it obtained a position estimation accuracy of 23 m.

## 1. Introduction

Solar-electrically powered fixed-wing unmanned aerial vehicles (UAVs) promise significantly increased flight endurance over pure electrically or even gas-powered aerial vehicles. Large-scale disaster relief, meteorological surveys in remote areas, and continuous border or wildlife protection benefit in particular from the multi-hour continuous flight capability provided by these robotic systems [1]. Traditional solar-powered UAVs require runways or complicated catapults for take-off and landing which are not commonly available in most field areas. The large size makes it inconvenient to apply and carry. The size of the UAV can be reduced to a wingspan of 2 to 5 m. With its excellent aerodynamic shape and lightweight structure, the hand-launched solar-powered UAV has great advantages in convenient application, long-endurance, and various mission execution [2,3,4].

Smaller-scale solar-powered UAVs are mostly designed for low-altitude long-endurance (LALE) applications [5]. Though faced with the more challenging meteorological phenomena of the lower atmosphere, low-altitude UAVs provide great potential of higher resolution imaging, reduced complexity and cost and simplified handling, for example, through hand-launched take-off. As a verification scheme, SoLong [3] has achieved a continuous 48-h flight using solar power and actively seeking out thermal updrafts, and Skysailor [4] has achieved a continuous 27-h solar-powered flight without the use of thermals, but both of these require a pilot to control them. AtlantikSolar [5] has completed an 81-h autonomous flight, except for launch and landing, which created the current unofficial flight endurance world record for all aircraft below 50 kg total mass. The current trend of the LALE solar-powered UAV is from the research stage to the real-life mission application and civil field [6].

The manufacturing cost is a crucial factor in the marketization of a UAV. It consists of body frame, energy system, engine system, flight controller, and payload. Except for the controller, the cost of other systems is relatively fixed. The industrial flight controller consists of high-precision sensors that ensure the UAV has sufficient control precision and trajectory tracking capability, but its high price accounts for a large proportion of the entire cost. In contrast, the flight controller consisting of a low-cost micro-electro-mechanical system (MEMS), inertial measurement unit (IMU), magnetometer, GPS and barometer has a great price advantage, but it lacks measurement accuracy and long-time reliability, and cannot be directly applied to a solar-powered UAV platform [7]. Therefore, it is a key technology to reduce the cost of UAV manufacturing by using a suitable state estimation algorithm for long endurance stable estimation of low-cost sensors [8]. The solar-powered UAV has the characteristics of low flight speed and long-endurance, but the measurement error of low-cost sensors will be further amplified with time accumulation and temperature changes. By using the appropriate state estimation algorithm, the measurement accuracy of the low-cost sensor is effectively improved to meet the trajectory tracking accuracy requirements [9], for example, a four-sided route with an area of 1 km^2^ has an acceptable tracking error of approximately 30 m (30 m/km^2^), and a heading error of nearly 13 degrees, thereby reducing the cost of the flight controller.

As a bridge between sensor measurement and controller calculation, the estimated state is fed back to the flight controller through the fusion of measurement of each sensor, and its accuracy will directly affect the positioning accuracy and control effect of the UAV. The nonlinearity and cross-axis sensitivity of industrial-grade sensors is significantly better than the low-cost sensors, as shown in Table 1. When the solar-powered UAV is equipped with a low-cost sensor, how to balance the error of the sensors for a long time to ensure the stable estimation of the state during the mission is the key to state estimation. With a large number of applications of low-cost flight controller platforms on miniature air vehicles, relevant research has also made some progress [10].

Dynamic observation theory and nonlinear filtering have proven to be effective in UAV state estimation. The complementary filter is a kind of data fusion algorithm based on frequency domain [11]. It uses gyroscope data in a short time, and regularly uses accelerometer and magnetometer data to correct the gyroscope. The Kalman filter (KF) is an optimal filter based on the state space method in the time domain, which resolves and eliminates noise according to the statistical characteristics of the noise measurement. For example, Extended Kalman filter (EKF), Unscent Kalman filter (UKF). The EKF method is the most widely used, and the core is to linearize the measurement model [12]. Different structures can be derived according to the type of sensor and the expression of the attitude angle. The UKF uses a deterministic approach to calculate the mean value and covariance, which results in approximations accurate to the third order (Taylor series expansions) [13]. However, the nonlinear method is computationally intensive and is not suitable for application on a low-cost flight controller.

Heikki [14] proposed a novel altitude estimation algorithm for fusing triaxial gyroscope and accelerometer measurements based on the EKF method. By introducing a variable covariance matrix and a low-cost, temperature-based calibration method, the robustness of the estimation results is improved. Michael [15] proposed an altitude determination system for small quadrotor UAV using consumer-grade Components-Off-The-Shelf (COTS). The exchange of estimated information is achieved by two extended Kalman filters without magnetometer. Nak [16] used an invariant extended Kalman filter (IEKF) for UAV navigation, based on EKF. Flight tests show that the method can be applied to rotary- and fixed-wing UAV platforms. Hu [17] proposed the external acceleration roll angle Kalman filter (EARAKF), which achieved a stable estimation of the roll angle during continuous overload with low-cost sensors.

In this paper, three novel state estimation algorithms are proposed to improve the measurement accuracy of low-cost sensors, reduce the cost of flight controller, and realize the application of three-stage series method in hand-launched solar-powered UAV. Additionally, the UAV is special, as it is designed without ailerons and a landing gear to ensure that the full wing is covered with solar cells and is lightweight (Figure 1). Since only the rudder is used for lateral control, the trajectory tracking accuracy of such a UAV is poor, and the state estimation algorithms can combine the characteristics of the UAV to achieve reliable heading estimation, ensuring that it meets the mission requirements and realizes the reduction of platform cost.

The paper is organized as follows. Low-cost sensors measurement error model and error statistics are described in Section 2. Section 3 presents three state estimation algorithms based on EKF. Section 4 verifies the above algorithms according to the whole mission path simulation. The field experiment results and model calibration methods are presented in Section 5, and Section 6 concludes the paper.

## 2. Sensors Measurement Error Model

The sensors combination in the low-cost flight controller is usually composed of MEMS IMU, magnetometer, pressure sensor and GPS [18], as shown in Figure 2. The SBG-Ellipse IMU module is chosen as an industrial-grade accuracy comparison.

The IMU consists of a triaxial gyroscope, a triaxial accelerometer, the hardware is MPU6050. The accelerometer with a capacitive transducer to convert the mass displacement into a voltage output. The measurements are subject to signal bias and random uncertainty, and the simulation model is constructed as follows:(1)$$\{\begin{array}{c} y_{accel,x}{=\dot{u}+qw-rv+gsin\theta +\beta }_{accel,x}{+\eta }_{accel,x}\\ y_{accel,y}=\dot{v}+ru-pw+gcos\theta sin\phi {+\beta }_{accel,y}{+\eta }_{accel,y}\\ y_{accel,z}=\dot{w}+pv-qv+gcos\theta sin\phi {+\beta }_{accel,z}{+\eta }_{accel,z} \end{array}$$where $\beta_{accel}$ is a bias term, which is dependent on temperature and should be calibrated prior to each flight. The $\eta_{accel}$ is zero-mean Gaussian noise with the variance $\sigma_{accel}^{2}$. The output of a gyroscope can be modeled as
(2)$$\gamma_{gyro}=k_{gyro}\Omega +\beta_{gyro}+\eta_{gyro}$$
where $\gamma_{gyro}$ corresponds to the measured angular rate in volts; $k_{gyro}$ is a gain converting the rate in radians per second to volts and Ω is the angular rate (these two items can be replaced by the real angular rate in the simulation); $\beta_{gyro}$ is a bias term, mainly dependent on temperature, and the drift is obviously greater than $\beta_{accel}$; $\eta_{gyro}$ is zero-mean Gaussian noise with covariance $\sigma_{gyro}^{2}$. The measurement model of the triaxial magnetometer is as follows:(3)$$\begin{array}{l} y_{mag}=\psi +\beta_{mag}+\eta_{mag}\\ B_{0}=R^{-1}\left(\phi ,\theta ,\psi \right){\left[\begin{array}{ccc} 0 & 0 & y_{mag} \end{array}\right]}^{T} \end{array}$$where β and η are similar to the above, and *R* is a direction cosine matrix. The pressure sensor is divided into absolute pressure sensor to measure altitude and differential pressure sensor to measure airspeed. The measurement models are
(4)$$y_{abspres}=\rho gh_{AGL}+\beta_{abspres}+\eta_{abspres}$$
(5)$$y_{diffpres}=\rho V_{a}^{2}/2+\beta_{diffpres}+\eta_{diffpres}$$

In the GPS measurement process, not only does the size of the positioning error need to be considered, but also the dynamic characteristics of the error are required. To model the transient behavior of the error, the Gauss–Markov process proved to be effective [19], and the expression is as follows:(6)$$v[n+1]=e^{-k_{GPS}T_{s}}v[n]+\eta_{GPS}[n]$$

The GPS measurement information includes UAV space position, ground speed and heading angle can be modeled as
(7)$$y_{GPS,n/e/h}[n]=p_{n/e/h}[n]+v_{n/e/h}[n]$$
(8)$$\begin{array}{l} V_{g}{}_{GPS}=\sqrt{{\left(V_{a}\cos\psi +\omega_{n}\right)}^{2}+{\left(V_{a}\sin\psi +\omega_{e}\right)}^{2}}+\eta_{V}\\ \chi_{GPS}=\tan^{-1}(V_{a}\sin\psi +\omega_{e},V_{a}\cos\psi +\omega_{n})+\eta_{\chi } \end{array}$$
where $\eta_{V}$ and $\eta_{\chi }$ are zero-mean Gaussian processes with variance $\sigma_{V_{g}}^{2}$ and $\sigma_{\chi }^{2}$, which can be obtained by using basic principles of uncertainty analysis:(9)$$\sigma_{V_{g}}=\sqrt{\frac{V_{n}^{2}\sigma_{V_{n}}^{2}+V_{e}^{2}\sigma_{V_{e}}^{2}}{V_{n}^{2}+V_{e}^{2}}}\;\sigma_{\chi }=\sqrt{\frac{V_{n}^{2}\sigma_{V_{e}}^{2}+V_{e}^{2}\sigma_{V_{n}}^{2}}{V_{n}^{2}+V_{e}^{2}}}$$

The statistical results of various low-cost sensor measurement errors are shown in Table 1, and the SBG-Ellipse is used for comparison.

According to the flight conditions of the UAV at low altitude, with an altitude of 600 m and an airspeed of 12.5 m/s during cruising conditions, combined with what from Table 1, the sensors are simulated as follows: No wind, an altitude of 600 m, an airspeed of 12.5 m/s, the controller temperature is 25 °C and remains unchanged, combined with the orthogonal error and bias of Table 1, the simulation results of various low-cost sensors are as follows. The solid line represents the true state and the dashed line represents the simulated measurement.

In Figure 3, (a,b), which is the IMU simulation, refers to Equations (1) and (2), where the accelerometer measurement is a high frequency signal; (c) is a magnetometer (refer to Equation (3)), and the measurement is a medium–low frequency signal. In order to consider the interference of the solar cell, the model error is appropriately increased, making the measurement slightly unstable; (d,e) is GPS (refer to Equations (6)–(8)); and (f) is pressure sensors simulation, as low frequency as GPS. The sensor modeling process mainly considers the influence of Gaussian white noise and bias, which is reflected in the range of variation range, frequency and steady-state deviation of the measured value. For the dynamic process, the measurement is consistent with the trend of the true signal. When entering the steady-state, the measurement noise is large and there is a significant deviation. 

## 3. EKF State Estimation Structure

The extended Kalman filter is similar to the linear Kalman filter, which uses an estimate of statistical variance to calculate the weight of sensor corrections to state estimates [20,21]. The state and observation equations of the nonlinear system are as follows:(10)$$\begin{array}{c} \dot{x}=f\left(x,u\right)+w\left(t\right)\\ z_{k}=h\left(x\left(t_{k}\right),u\right){+v}_{k} \end{array}$$where w(t) is used to account for disturbance and unmodeled part, $v_{k}$ represents the measurement noise. It is assumed that the w(t) and $v_{k}$ are zero-mean Gaussian noise, and covariance *Q* and *R*, respectively. The EKF is first-order linearization near the state, and the system time update equations are
(11)$$\dot{\hat{x}}=f\left(\hat{x},u\right)$$
(12)$$A\left(x,u\right)=\frac{\partial f\left(x,u\right)}{\partial x}|{}_{x=\hat{x}}$$
(13)$$\dot{P}=A\left(\hat{x},u\right)P+\mathrm{PA}{\left(\hat{x},u\right)}^{T}+Q$$

For the update of state estimates when a measurement is received, the measurement update equations are
(14)$$C\left(x,u\right)=\frac{\partial h\left(x^{-},u\right)}{\partial x}$$
(15)$$L=P^{-}C^{T}{\left(R+C\left(\hat{x},u\right)PC{\left(\hat{x},u\right)}^{T}\right)}^{-1}$$
(16)$$P=\left(I-LC\left(\hat{x},u\right)\right)P^{-}$$
(17)$$\hat{x}={\hat{x}}^{-}+L\left(z-h\left({\hat{x}}^{-},u\right)\right)$$
where **A** is the linearized state update matrix, **C** is the linearized model output matrix, **P** is the state covariance matrix, **L** is the Kalman gain matrix, **Q** is the process noise covariance matrix, and **R** is the sensor covariance matrix.

Based on the EKF, this paper divides the state estimation algorithm into full-state estimation and three-state series estimation method according to whether the estimation structure is hierarchical. The full-state estimation consists of the direct and indirect method. The same hand-launched UAV simulation model was used, with an altitude of 600 m and an airspeed of 12.5 m/s, and the hovering condition of the windless conditions.

### 3.1. Three-Stage Series State Estimation

This algorithm divides the estimation process into three stages. Firstly, the measurement data are low-pass filtered, and gyroscope data are used for prediction and combined with the accelerometer for measurement update, which is the altitude estimation. The second stage is the heading estimation, especially when the UAV is flying in the wind field. The accurate heading angle is crucial for stable control. The final stage is the navigation estimation, based on the previous results and fused with GPS for measurement updates. The algorithm structure is shown in Figure 4.

#### 3.1.1. Attitude Estimation

For this stage, the states, inputs and outputs are
$$x={\left[\begin{array}{cc} \phi & \theta \end{array}\right]}^{T}\;u={\left[\begin{array}{ccccc} p & q & r & V_{a} & H \end{array}\right]}^{T}\;z={\left[\begin{array}{ccc} a_{x} & a_{y} & a_{z} \end{array}\right]}^{T}$$

The state variables to be estimated in this stage are pitch and roll. Angular rates, airspeed and altitude make up the input vector, **u**, and body frame accelerations composed the system outputs, **z**, for this stage. According to the relationship between body frame rotations and changes in roll and pitch angle [22], the time update of the states is
(18)$$\left[\begin{array}{c} \dot{\hat{\phi }}\\ \dot{\hat{\theta }} \end{array}\right]=f\left(\hat{x},u\right)=\left[\begin{array}{c} p+qsin\hat{\phi }tan\hat{\theta }+rcos\hat{\phi }tan\hat{\theta }\\ qcos\hat{\phi }-rsin\hat{\phi } \end{array}\right]$$

After evaluating the linearization about the current estimate states, the linearized state update matrix **A** is obtained:(19)$$A\left(x,u\right)=\frac{\partial f\left(x,u\right)}{\partial x}|{}_{x=\hat{x}}=\left[\begin{array}{cc} q\cos\hat{\phi }\tan\hat{\theta }-r\sin\hat{\phi }\tan\hat{\theta } & \frac{q\sin\hat{\phi }-r\cos\hat{\phi }}{\cos^{2}\hat{\theta }}\\ -q\sin\hat{\phi }-r\cos\hat{\phi } & 0 \end{array}\right]$$

According to Equation (1), assuming that linear acceleration is zero, that is, $\dot{u}=\dot{v}=\dot{w}\approx 0$, the measurement update and linearized model output matrix **C** are shown as follows.
(20)$$\left[\begin{array}{c} {\hat{a}}_{x}\\ {\hat{a}}_{y}\\ {\hat{a}}_{z} \end{array}\right]=h\left(\hat{x},u\right)=\left[\begin{array}{c} qV_{a}\sin\hat{\theta }+g\sin\hat{\theta }\\ rV_{a}\cos\hat{\theta }-pV_{a}\sin\hat{\theta }-g\cos\hat{\theta }\sin\hat{\phi }\\ -qV_{a}\cos\hat{\theta }-g\cos\hat{\theta }\sin\hat{\phi } \end{array}\right]$$
(21)$$C\left(x,u\right)=\frac{\partial h\left(x,u\right)}{\partial x}|{}_{x=\hat{x}}=\left[\begin{array}{cc} 0 & qV_{a}\cos\hat{\theta }+g\cos\hat{\theta }\\ -g\cos\hat{\phi }\cos\hat{\theta } & -rV_{a}\sin\hat{\theta }-pV_{a}\cos\hat{\theta }+g\sin\hat{\phi }\sin\hat{\theta }\\ g\sin\hat{\phi }\cos\hat{\theta } & (qV_{a}+g\cos\hat{\phi })\sin\hat{\theta } \end{array}\right]$$

Using Equations (11) to (17), the state covariance and Kalman gain matrix are updated to achieve the altitude angle estimate. The simulation results of this stage are as follows.

As shown in Figure 5, the subscribe hat and true are the estimated and real state, respectively. Using the variable gain observer (VGO) as a comparison, the three-stage series algorithm has higher accuracy for altitude estimation. The noise of the gyroscope is reflected in the time update, while the accelerometer noise is applied to the measurement update, and the estimated altitude is used as the input of the heading estimation.

#### 3.1.2. Heading Estimation

An accurate heading is important to the lateral control of a UAV without ailerons. The rudder maneuvering produces a yaw moment, then the side slip occurs, and the UAV rolls depending on its own stability. Since the roll control is not direct, it will affect the trajectory tracking accuracy. Magnetometer update heading estimation is useful for cases where a wind induced crab needs to be accounted for when interpreting data from cameras or other sensors [23]. Due to the interference of the solar cell, the magnetometer can be externally mounted and installed at the wing tip to obtain accurate measurement, which is suitable for large aspect ratio solar-powered UAV. The groundspeed will reduce under windy conditions and the heading approximated from GPS will become inaccurate, leading to the vehicle flying uncontrollably downwind [24]. In this stage, the heading estimation is a single-state EKF. The state, inputs and outputs are
$$x=\psi \;u={\left[\begin{array}{cccc} \phi & \theta & q & r \end{array}\right]}^{T}\;z={\left[\begin{array}{ccc} B_{x} & B_{y} & B_{z} \end{array}\right]}^{T}$$

The time update of the estimation using gyroscope measurements by the expression
(22)$$\dot{\hat{\psi }}=f\left(\hat{x},u\right)=q\frac{sin\phi }{cos\theta }+r\frac{cos\phi }{sin\theta }$$
(23)$$A\left(x,u\right)=\frac{\partial f\left(x,u\right)}{\partial x}|{}_{x=\hat{x}}=0$$

As the single-state linearization yields a zero matrix, only model noise is considered in the covariance matrix update process. The sensor model is simply a rotational transformation of the earth-frame magnetic field to the body-frame sensor, given by
(24)$$\left[\begin{array}{c} {\hat{B}}_{x}\\ {\hat{B}}_{y}\\ {\hat{B}}_{z} \end{array}\right]=h(\hat{x},u)=\left[\begin{array}{ccc} \cos\theta \cos\hat{\psi } & \cos\theta \sin\hat{\psi } & -\sin\theta \\ \sin\phi \sin\theta \cos\hat{\psi }-\cos\phi \sin\hat{\psi } & \sin\phi \sin\theta \sin+\cos\phi \cos\hat{\psi } & \sin\phi \cos\theta \\ \cos\phi \sin\theta \cos\hat{\psi }+\sin\phi \sin\hat{\psi } & \cos\phi \sin\theta \sin-\sin\phi \cos\hat{\psi } & \cos\phi \cos\theta \end{array}\right]\left[\begin{array}{c} B_{0_{x}}\\ B_{0_{y}}\\ B_{0_{z}} \end{array}\right]$$

Linearizing the current estimates, the linearized model output matrix is
(25)$$C\left(x,u\right)=\frac{\partial h\left(x,u\right)}{\partial x}|{}_{x=\hat{x}}=\left[\begin{array}{c} -\cos\theta \sin\hat{\psi }B_{0_{x}}+\cos\theta \cos\hat{\psi }B_{0_{y}}\\ (-\sin\phi \sin\theta \sin\hat{\psi }-\cos\phi \cos\hat{\psi })B_{0_{x}}+(\sin\phi \sin\theta \sin\hat{\psi }+\cos\phi \sin\hat{\psi })B_{0_{y}}\\ (-\sin\phi \sin\theta \sin\hat{\psi }-\cos\phi \cos\hat{\psi })B_{0_{x}}+(\sin\phi \sin\theta \sin\hat{\psi }-\cos\phi \sin\hat{\psi })B_{0_{y}} \end{array}\right]$$

The heading estimation simulation is performed using the UAV hovering state, as shown in Figure 6. As the VGO does not have a separate heading estimate, the simulation results are compared with the real and estimated yaw angle.

Since the solar cell will affect the measurement of the magnetometer, the bias and noise are more obvious, as shown in Figure 3c. Accurate attitude estimation can ensure the accuracy of heading prediction. The measurement update of the magnetometer makes the trend of estimation consistent with the real state. The simulation result has a delay of 1 s and an error of 2 degrees. 

#### 3.1.3. Navigation Estimation

The final stage of this state estimation scheme keeps an estimate of vehicle location and path angle. With the altitude and heading obtained in first two stages and measured airspeed, the location, groundspeed and course angle of the airplane are predicted using a simple kinematic model and check against GPS measurements [25]. If the flight path angle γ=0, then the evolution of the position is given by
(26)$$\begin{array}{c} {\dot{P}}_{N}{=V}_{a}{cos\psi cos\theta -W}_{N}\\ {\dot{P}}_{E}{=V}_{a}{sin\psi cos\theta -W}_{E} \end{array}$$

By differentiating Equation (8), the evolution of the ground speed is given by
(27)$${\dot{V}}_{g}=\frac{\left(V_{a}{cos\psi +W}_{N}\right)\left({-V}_{a}\dot{\psi }sin\psi \right)+\left(V_{a}{sin\psi +W}_{E}\right)\left(V_{a}\dot{\psi }cos\psi \right)}{V_{g}}$$

According to the kinematic equation of the aircraft during the coordinated turning process, the evolution of χ is given by
(28)$$\dot{\chi }=\frac{g}{V_{g}}tan\phi cos\left(\chi -\psi \right)$$

The state, inputs and output of this stage is
$$\begin{array}{l} x={\left[\begin{array}{cccccc} P_{N} & P_{E} & V_{g} & \chi & W_{N} & W_{E} \end{array}\right]}^{T}\\ u={\left[\begin{array}{cccccc} V_{a} & q & r & \phi & \theta & \psi \end{array}\right]}^{T}\\ z={\left[\begin{array}{cccccc} y_{GPS,N} & y_{GPS,E} & y_{GPS,Vg} & y_{GPS,\chi } & y_{W_{N}} & y_{W_{E}} \end{array}\right]}^{T} \end{array}$$

The nonlinear propagation and linearized state update matrix are given by
(29)$$\left[\begin{array}{c} {\hat{P}}_{N}\\ {\hat{P}}_{E}\\ \begin{array}{c} {\;\hat{V}}_{g}\\ \hat{\chi }\\ \begin{array}{c} {\;\hat{W}}_{N}\\ {\;\hat{W}}_{E} \end{array} \end{array} \end{array}\right]=f\left(\hat{x},u\right)=\left[\begin{array}{c} V_{a}{cos\psi cos\theta -\hat{W}}_{N}\\ V_{a}{sin\psi cos\theta -\hat{W}}_{E}\\ \begin{array}{c} \left[\left(V_{a}{cos\psi +\;\hat{W}}_{N}\right)\left({-V}_{a}\;\dot{\psi }sin\psi \right)+\left(V_{a}{sin\psi +\;\hat{W}}_{E}\right)\left(V_{a}\dot{\psi }cos\psi \right)\right]{/\;\hat{V}}_{g}\\ gtan\phi cos\left(\hat{\chi }-\psi \right){/V}_{g}\\ 0_{2\times 1} \end{array} \end{array}\right]$$
(30)$$\dot{f}=\left[\begin{array}{cc} 0_{4\times 2} & \begin{array}{ccc} \begin{array}{c} cos\hat{\chi }\\ sin\hat{\chi }\\ \begin{array}{c} -{\dot{\hat{V}}}_{g}/{\hat{V}}_{g}\\ -gtan\phi cos\left(\hat{\chi }-\psi \right)/{\hat{V}}_{g}^{2} \end{array} \end{array} & \begin{array}{c} -V_{g}sin\hat{\chi }\\ V_{g}cos\hat{\chi }\\ \begin{array}{c} 0\\ -gtan\phi cos\left(\hat{\chi }-\psi \right)/{\hat{V}}_{g} \end{array} \end{array} & \begin{array}{cc} \begin{array}{c} -1\\ 0\\ \begin{array}{c} -V_{a}\dot{\psi }sin\psi \\ 0 \end{array} \end{array} & \begin{array}{c} 0\\ -1\\ \begin{array}{c} V_{a}\dot{\psi }cos\psi \\ V_{a}\dot{\psi }cos\psi \end{array} \end{array} \end{array} \end{array}\\ 0_{2\times 2} & 0_{2\times 4} \end{array}\right]$$

GPS measurement information includes position, ground speed and course angle. In order to estimate the wind field, a pseudo measurement of wind speed is constructed as follows:(31)$$\begin{array}{l} y_{W_{N}}=V_{a}\cos\psi +W_{N}-V_{g}\cos\chi \\ y_{W_{E}}=V_{a}\sin\psi +W_{E}-V_{g}\sin\chi \end{array}$$

Therefore, the measurement update and the Jacobian matrix are given by
(32)$$h(\hat{x},u)={\left[\begin{array}{cccccc} {\hat{P}}_{N} & {\hat{P}}_{E} & {\hat{V}}_{g} & \hat{\chi } & V_{a}\cos\psi +{\hat{W}}_{N}-{\hat{V}}_{g}\cos\hat{\chi } & V_{a}\sin\psi +{\hat{W}}_{E}-{\hat{V}}_{g}\sin\hat{\chi } \end{array}\right]}^{T}$$
(33)$$C\left(x,u\right)=\frac{\partial h\left(x,u\right)}{\partial x}{|}_{x=\hat{x}}=\left[\begin{array}{cc} I_{4} & 0_{4\times 2}\\ \begin{array}{cc} 0_{2\times 4} & \begin{array}{cc} -cos\chi & {\dot{V}}_{g}sin\chi \\ -sin\chi & {-\;\dot{V}}_{g}cos\chi \end{array} \end{array} & I_{2} \end{array}\right]$$

The simulation results of the navigation estimation are shown in Figure 7. The estimation results of ground speed and wind field are better than the VGO algorithm; the groundspeed static bias is 0.5 m/s, which is improved to less than 0.2 m/s. The estimated GPS course is based on the results of the first two stages, so the position accuracy is higher. Algorithm 1 summary of the three-stage series estimation method.

**Algorithm 1.** Three-stage series state estimationStage 1: Attitude estimation1: Low-pass filtering of barometers, airspeed sensor and gyroscope measurements, set initial x^.2: Update the roll and pitch altitude, Equation (18).3: Compute the linearized state update, covariance matrix and update the prediction, Equations (13) and (19).4: **for** each accelerometer axis **then**5:  Calculate appropriate sensor output, linearized output, and Kalman gain matrix, Equations (15), (20) and (21).6:  Update state estimate and covariance matrix, Equations (16) and (17).7: **end for**Stage 2:Heading estimation1: Obtain q and r from gyroscope update and ϕ and θ from stage 1.2: Update the heading angle using kinematic equation, Equation (22).3: Compute the linearized state update matrix, zero matrix, and covariance matrix, Equations (13) and (23).4: **if** magnetometer measurement available **then**5:  **for** each magnetometer axis do6:   The same process as the stage 1, sensor measurement update is different, Equations (24) and (25).7:  **end for**8: **end if**Stage 3:Navigation estimation1: The airspeed is updated by the airspeed sensor, assign initial value to the altitude angle.2: Update the state estimate using kinematic equation Equation (29)3: Compute the linearized state update and covariance matrix, Equations (13) and (30)4: **if** GPS measurement available **then**5:  **for** each measurement state **do**6:   The same process as the stage 1, sensor measurement update is different, Equations (32) and (33).7:  **end for**8: **end if**

### 3.2. Full-State Direct State Estimation

The hierarchical method is used to simplify the model, reduce the order of the matrix, ignore the mutual influence of the state parameters, and obtain a reliable heading estimation. This reduces the amount of calculation necessary but the accuracy is limited. The full-state method treats all parameters as a whole, the measurement error of IMU is used as a time update, and other sensors are used as measurement update to get the complete state.

When the mission is sensitive to airspeed and altitude, and the flight controller has feasible computing power, full-state direct state estimation is a suitable method to apply. The algorithm structure is shown in Figure 8. Due to the different sampling frequencies, the corresponding state of the fast sensor measurement update using the air pressure sensor includes altitude, airspeed and side slip angle pseudo measurement [26]. The states corresponding to the slow update using GPS include location, ground speed and course.

The state, measurement input of the time update process are $$x={\left(P^{T},V^{T},\theta^{T},b^{T},w^{T}\right)}^{T}\;y={\left(y_{accel}^{T},y_{gyro}^{T}\right)}^{T}$$


Using the gyro and accelerometer models, the equations of motion can be written as (34)$$\{\begin{array}{c} \dot{P}=R\left(\theta \right)V\\ {\dot{V}=V}^{\times }\left(y_{\mathrm{gyro}}{-b-\eta }_{\mathrm{gyro}}\right)+\left(y_{\mathrm{accel}}{-b-\eta }_{\mathrm{accel}}\right)+\eta \\ \begin{array}{c} \dot{\Omega }=S\left(\Theta \right)\left(y_{\mathrm{gyro}}{-b-\eta }_{\mathrm{gyro}}\right)\\ \begin{array}{c} {\;\dot{b}=0}_{3\times 1}\\ {\;\dot{w}=0}_{2\times 1} \end{array} \end{array} \end{array}$$
where $P={\left(p_{n},p_{e},p_{d}\right)}^{T}$, $V={\left(u,v,w\right)}^{T}$, $\theta ={\left(\phi ,\theta ,\psi \right)}^{T}$, $\omega ={\left(p,q,r\right)}^{T}$, b is the bias vector, η is the noise vector, w is the wind filed vector, R(θ) is the direction cosine matrix, and S(θ) is the transformation matrix of the angular velocity of body axes to inertial axes. Therefore, the state propagation equation is (35)$$\dot{x}=f\left(x,y\right){+G}_{g}\left(x\right)\eta_{\mathrm{gyro}}{+G}_{a}\left(x\right)\eta_{\mathrm{accel}}+\eta$$


The time update model of the state and the Jacobian matrices are as follows:(36)$$f\left(x,y\right)=\left(\begin{array}{c} R\left(\theta \right)V\\ V^{\times }\left(y_{gyro}-b\right)+\left(y_{accel}-b\right)+R^{T}\left(\theta \right)g\\ S\left(\theta \right)\left(y_{gyro}-b\right)\\ 0_{3\times 1}\\ 0_{2\times 1} \end{array}\right)$$(37)$$A\left(x,y\right)=\frac{\partial f}{\partial x}=\left(\begin{array}{ccccc} 0_{3\times 3} & R\left(\theta \right) & \frac{\partial \left[R\left(\theta \right)V\right]}{\partial \theta } & 0_{3\times 3} & 0_{3\times 2}\\ 0_{3\times 3} & -{\left(y_{gyro}-b\right)}^{\times } & \frac{\partial \left[R^{T}\left(\theta \right)g\right]}{\partial \theta } & -V^{\times } & 0_{3\times 2}\\ 0_{3\times 3} & 0_{3\times 3} & \frac{\partial \left[S\left(\theta \right)\left(y_{gyro}-b\right)\right]}{\partial \theta } & -S\left(\theta \right) & 0_{3\times 2}\\ 0_{3\times 3} & 0_{3\times 3} & 0_{3\times 3} & 0_{3\times 3} & 0_{3\times 2}\\ 0_{2\times 3} & 0_{2\times 3} & 0_{2\times 3} & 0_{2\times 3} & 0_{3\times 2} \end{array}\right)$$

The fast sensor measurement update includes three parts, static and differential pressure sensor and pseudo side-slip angle. According to Equations (4) and (5), the measurement model and associated Jacobian matrices are given by (38)$$h_{static}\left(x\right)=-\rho gp_{d}\;h_{diff}\left(x\right)=\frac{1}{2}\rho V_{a}^{2}=\frac{1}{2}\rho {\left(V-R^{T}\left(\theta \right)w\right)}^{T}\left(V-R^{T}\left(\theta \right)w\right)$$
(39)$$C_{static}\left(x\right)=\left(0,0,-\rho g,0\ldots \right)\;C_{diff}\left(x\right)=\left(0_{3,1},V-R^{T}\left(\theta \right)w,\frac{\partial h}{\partial \theta },0_{3,1},\frac{\partial h}{\partial w}\right)$$


With the given sensors, the side-slip angle or side-to-side velocity is unobservable and drift occurs, but it is important when the control system needs to obtain reliable lateral information to control the flight path [12]. To correct this parameter, a pseudo measurement is proposed on the side-slip angle by assuming that it is zero. Therefore, the model for the pseudo-sensor is given by (40)$$h_{\beta }=v_{r}\left(V,\theta ,w\right)=\left[\begin{array}{ccc} 0 & 1 & 0 \end{array}\right]\left(V-R^{T}\left(\theta \right)w\right)$$


The associate Jacobian matrix is given by (41)$$C_{\beta }\left(x\right)=\left(0_{1\times 3},\frac{\partial h}{\partial V},\frac{\partial h}{\partial \theta },0_{1\times 3},\frac{\partial h}{\partial w}\right)$$


The GPS measurement is updated at a low frequency, including north and east position, ground speed and heading. Similarly, the measurement model and the corresponding Jacobian matrices are as follows:(42)$$\begin{array}{ll} h_{GPS,n}\left(x\right)=p_{n} & h_{GPS,e}\left(x\right)=p_{e}\\ h_{GPS,Vg}\left(x\right)=\left\|[I_{2,2},0_{2,1}]R\left(\theta \right)V\right\| & h_{GPS,\chi }\left(x\right)=\tan^{-1}\left(V_{g,e}/V_{g,n}\right) \end{array}$$
(43)$$\begin{array}{ll} C_{GPS,n}\left(x\right)=\left(1,0,0\ldots \right) & C_{GPS,e}\left(x\right)=\left(0,1,0\ldots \right)\\ C_{GPS,Vg}\left(x\right)=\left(0,V^{T}R^{T}\left(\theta \right)P^{T}PR\left(\theta \right),0,0\right) & C_{GPS,\chi }\left(x\right)=\left(0,\frac{\partial h}{\partial V},\frac{\partial h}{\partial \theta },0,0\right) \end{array}$$


The comparison between the real and the estimate state of all parameters is shown in Figure 9. Algorithm 2 is a summary of full-state direct state estimation method.

Compared with Figure 5 and Figure 6, the accuracy of full-state method is obviously superior in position, speed, and heading. When entering the steady state, the altitude deviation is small, the error precision is less than 8%, the GPS-related state measurement accuracy is higher, and the position error precision is approximately 3 m, ensuring sufficient trajectory tracking accuracy. Due to the increase of accuracy of the estimated ground speed and course, this method is suitable for reliable trajectory tasks or fixed-area hovering, such as animal photography or precise takeoff and landing.

**Algorithm 2.** Full-state direct state estimationTime update1: Obtain ω and a from gyroscope and accelerometer, set initial $\hat{x}$.2: Update the state estimate using the kinematic equation, Equation (35).3: Compute the linearized state update matrix and covariance variable, Equations (13) and (37).Fast sensor update1: **if** air pressure sensor measurement available, **then**2:  **for** static and differential pressure sensor measurement **do**3:   Calculate sensor output, linearized output equations, and Kalman gain, Equations (15), (38) and (39).5:   Update state estimate and covariance, Equations (16) and (17)6:  **end for**7:  Calculate pseudo side slip angle output and linearized output equations, Equations (40) and (41).8:  Using zero to update state estimate, Equations (15)–(17)9: **end if**Slow sensor update1: **if** new GPS measurement available, **then**2:  **for** each measurement state **do**3:   Calculate location, groundspeed and course output and linearized output equations, Equations (42) and (43)4:   Using measurement to update state estimate, Equations (15)–(17)5:  **end for**6: **end if**

### 3.3. Full-State Indirect State Estimation

The direct state estimate method focuses on the observation of all measurement states, while for the indirect estimation method the idea is to filter the error states, which should satisfy the linear and Gaussian assumptions better than the direct implementation. The biggest difference of indirect method is that the error estimation vector is introduced, which is predicted by time update, and the estimation of states is achieved by observation and compensation of errors. Let **x** be the true state, $\hat{x}$ the estimated state, then $\tilde{x}=x-\hat{x}$ is the error state. Similar to the previous section, if $\;\hat{x}$ satisfies (44)$$\dot{\hat{x}}=f\left(\hat{x},y\right)$$


The state error evolves according to Equation (35), $\dot{\tilde{x}}$ can be written as (45)$$\dot{\hat{x}}=f\left(\hat{x},y\right)+G\left(x\right)\eta_{s}{+\eta }_{i}-f\left(\hat{x},y\right)$$


Using the Tylor series expansion up to the linear term, **f** and **G** are equivalent to (46)$$\begin{array}{c} f\left(x,y\right)\approx f\left(\hat{x},y\right)+A\left(\hat{x},y\right)\tilde{x}\\ G\left(x\right)\eta_{s}\approx G\left(x\right)\eta_{s} \end{array}$$


Equation (45) can be simplified as (47)$$\dot{\tilde{x}}=A\left(\hat{x},y\right)+G\left(\hat{x}\right)\eta_{s}{+\eta }_{i}$$


The basic idea is to run the indirect filter on the state error, then to correct the state estimate at each step by adding in the error, and the state error is reset to zero after each correction. In the time update process, the state error propagate equation can be written as (48)$$\dot{\tilde{x}}=A\left(\hat{x},y\right)\tilde{x}$$


In the measurement update process, it is also necessary to introduce a measurement of the state error, and Equation (17) can be rewritten as (49)$$\begin{array}{c} {\tilde{x}}^{+}{=\tilde{x}}^{-}+L\left(z-h\left({\hat{x}}^{-},u\right)-C\left({\hat{x}}^{-},u\right){\tilde{x}}^{-}\right)\\ {\hat{x}}^{-}{=\hat{x}}^{+}{+\tilde{x}}^{+} \end{array}$$


The state error is set to zero after each estimation, that is $\tilde{x}=0$. The algorithm is outlined in Figure 10.

As the supplement to the direct filter, it is more computationally intensive and more complex, but the elimination of measurement bias is better than the first two methods. Due to the large size of the state matrix, the computation power of low-cost flight controller is a challenge. Figure 11 shows the simulation results of the indirect method, Table 2 compares the accuracy of the different algorithms, and Algorithm 3 is a summary of the method.

According to the above, it can be seen that the estimation accuracy of indirect method is better than that of direct method, and the estimation accuracy of wind field is higher. The indirect method has further improved the estimation accuracy of the altitude and position. Compared with Figure 9, the altitude deviation is small at the beginning, the deviation after steady state is reduced, and the result is more stable. Therefore, this method is suitable for large-scale solar-powered UAV or mission with high trajectory tracking accuracy.

**Algorithm 3.** Full-state indirect state estimationTime update1: Obtain ω and a from gyroscope and accelerometer, set initial $\hat{x}$ and $\tilde{x}=0$.2: Update the state estimate using the kinematic equation and linearized state, Equations (35) and (37).3: Propagate $\tilde{x}$ and P according to Equation (Equations (13) and (48))Fast sensor update1: **if** air pressure sensor measurement available **then**2:  **for** static and differential pressure sensor measurement **do**3:   Calculate sensor output, linearized output equations, and Kalman gain, Equations (15), (38) and (39).4:   Update state covariance, state error and estimate state, Equations (16) and (49).5:  **end for**6:  Calculate pseudo side slip angle output and linearized output equations, Equations (40) and (41).7:  Using zero to update state estimate, Equations (15), (16) and (49)8: **end if**Slow sensor update1: **if** new GPS measurement available **then**2:  **for** each measurement state **do**3:   Calculate location, groundspeed and course output and linearized output equations (Equations (42) and (43)4:   Using measurement to update state error (Equations (15), (16) and (49))5:  **end for**6: **end if**7: State error reset zero, $\tilde{x}=0$.

## 4. Simulation

In order to verify the effectiveness of the proposed algorithm, the simulation of the complete mission path is carried out for the same UAV simulation model. A typical clime-cruise-hover-descend is used as the mission path, and the hovering process performs specific tasks, and the estimation accuracy in each flight phase and state switching is obtained. The flight trajectory and estimated parameters are shown in Figure 12 and Figure 13.

All three algorithms can achieve autonomous flight in the mission process, and the altitude and trajectory accuracy are different. As the static pressure sensor and airspeed sensor are only processed by low-pass filtering, the static bias of three-stage series method will appear in the estimation of altitude and airspeed, which is 5.3 m and 0.9 m/s, respectively, and the error of flight trajectory will also appear in the heading estimation, which is approximately 10 degrees, resulting in a position error of 6 m in the hover phase. 

Full-state estimation method fuses the measured data of pressure sensor and GPS, and has higher accuracy of position and velocity estimation, and there is only a small error in altitude, airspeed, and groundspeed. Full-state methods can consider the mutual influence between estimated states, and the parameter is not abruptly changed during the state switching, which is obvious in the three-stage series method. The altitude and position accuracy of the direct method are 1.2 degrees and 3.4 m, respectively. The indirect method achieves similar precision, but the estimated result is more stable. Since this method is based on the elimination of state error, the altitude estimation accuracy is high and the steady-state process estimation accuracy fluctuation is small.

Table 3 compares the three algorithms from five aspects: Algorithm complexity, hardware requirements, number of estimable parameters, estimation accuracy and application. The indirect full-state algorithm has the highest accuracy, but the hardware requirements are also highest. The direct full-state algorithm is more compromised, and the three-stage series has the adequate heading accuracy, with the lowest controller computation requirements and simplest complexity, but the estimated parameters are limited.

## 5. Field Experiment

A full-scale electronic UAV, with a wingspan of 3 m and a weight of 3 kg without solar cells, was equipped with a low-cost flight controller and the three-stage series estimation algorithm were loaded for the full mission flight verification (Figure 14). The cost of the flight controller is less than $3000, and the whole system is less than $8000. The experimental site is located at (108°56′3.07″ E, 34°58′58.85″ N), with an altitude of 630 m, an average wind speed of 3.5 m/s, a ground temperature of 10 °C, and during autumn.

The UAV has no landing gear and it takes off through hand-launch from the car. The fuselage touches the ground when landing. The predicted state during the calculation is the altitude, position, and groundspeed. In the Figure 15a,c are three-dimensional and two-dimensional flight trajectories, respectively, and (b) is parameters and commands recorded by flight controller. The directly measured states include altitude, airspeed, longitude and latitude; the IMU data are shown in the dotted line in Figure 16. Through the control of pitch and roll angle, altitude and airspeed control results, the three-stage series state estimation algorithm can provide a reliable state for flight controller. The calculation frequency is 50 Hz, and the IMU is also sampled based on the same frequency; the GPS measurement frequency is 10 Hz, which is the same as the data transmission frequency; and the data link frequency band is 915 MHz. The variation range is 8 m, the airspeed is 9–15 m/s, the groundspeed is 6–16 m/s, the roll angle accuracy is 6.5 degrees, the pitch angle is 3 degrees, the yaw angle is 9.3 degrees, and the trajectory tracking precision is nearly 23 m.

The field flight test can not only verify the feasibility of the state estimation algorithm, but also calibrate some parameters in the modeling process, such as sensor measurement bias, noise and covariance matrix **R**. The Finite Impulse Response (FIR) filter is used to filter the flight data [27], as shown in Figure 16. The measurement result of the IMU is regarded as the raw data, and the filtered parameter is regarded as the real state. The bias and noise of the sensor model can be obtained by the following equations:(50)$$\begin{array}{l} \eta_{correct}=\frac{{X_{m}|}_{\min}^{\max}-{X_{f}|}_{\min}^{\max}}{2}\\ \beta_{correct}=E\left(X_{m}\right)-E\left(X_{f}\right) \end{array}$$where $\eta_{correct}$ is the corrected measurement noise and $\beta_{correct}$ is the corrected measurement bias. The subscripts *m* and *f* represent measured and filtered data.

The statistical data are of the measured noise and deviation of the IMU is shown in Table 4. For the noise of the accelerometer and the bias of the gyroscope, the value in the *y* direction is smaller than the *x* and *z* directions. The calibration of the covariance matrix **R** in the state estimation process is shown as follows:(51)$$R_{accel}=\left[\begin{array}{ccc} {\left(0.0818\right)}^{2} & 0 & 0\\ 0 & {\left(0.0193\right)}^{2} & 0\\ 0 & 0 & {\left(0.251\right)}^{2} \end{array}\right]\;R_{gyro}=\left[\begin{array}{ccc} {\left(0.0723\right)}^{2} & 0 & 0\\ 0 & {\left(0.0906\right)}^{2} & 0\\ 0 & 0 & {\left(0.0898\right)}^{2} \end{array}\right]$$

For the velocity vector, $V={\left(u,v,w\right)}^{T}$, both *u* and *w* are greater than *v* during the mission process. A penalty factor τ can be introduced in the matrix **R***_accel_* to get a more general expression, rewritten as
(52)$$R_{accel}={\left[\begin{array}{ccc} R_{x_{0}} & \tau R_{y_{0}} & R_{z_{0}} \end{array}\right]}^{T}$$

## 6. Conclusions

This paper begins with the cost of the flight controller of a hand-launched solar-powered UAV without ailerons and landing gear, and proposes three novel state-estimation algorithms according to whether they are hierarchical, direct or indirect. The UAV trajectory tracking accuracy is improved by three algorithms and the platform cost is reduced.

Firstly, the measurement process of the low-cost sensors is modeled, and the simulation of the model is completed based on the statistical values of the measurement error and bias. The measurement of the accelerometer is high frequency, the gyroscope and the magnetometer are in middle frequency, and the GPS measurement is low frequency.

The three-stage series method reduces the longitudinal accuracy, and focuses on the estimation of heading angle to improve the trajectory precision, which is simple, with a clear hierarchy, limited estimation parameters, and is suitable for application on low-cost small UAV platforms. The full-state method can also obtain side-slip angle or side-to-side velocity estimation to improve the control precision of the aileron free UAV. For the solar-powered UAV, a method of fixing the magnetometer externally can be selected to ensure heading accuracy. The full-state direct method has a higher estimation accuracy and more estimation parameters, but the increase of the algorithm calculation requires a higher-level controller, suitable for high control precision and harsh takeoff and landing conditions. The indirect method has the highest accuracy estimation and the highest requirements for flight controllers, which can be applied to large solar-powered UAV platforms.

Simulations are presented to validate the proposed methods, and the results show the three-stage series method has a position accuracy of approximately 6 m, while the direct and indirect methods are both 3.4 m; however, the stability and state switching of the indirect method are better. Afterwards, the three-stage series algorithm is loaded on the full-scale low-cost electric UAV. The results illustrate that the three-stage series algorithm can provide a reliable state for the controller and achieve stable control, the yaw angle precision is 9.3 degrees, and the trajectory tracking precision is nearly 23 m. Finally, the sensor model and covariance matrix **R** corrected according to the recorded and filtered data of the IMU during flight. 

As a next step, we will conduct research on a long-endurance, high-reliable flight control system, and adaptability to harsh environmental condition. With the long time reliability test of the hardware, the service life of the low-cost solar-powered UAV can be obtained. The completion of this series of experiments makes it closer to civilian applications.

## Figures and Tables

**Figure 1 sensors-19-04627-f001:**
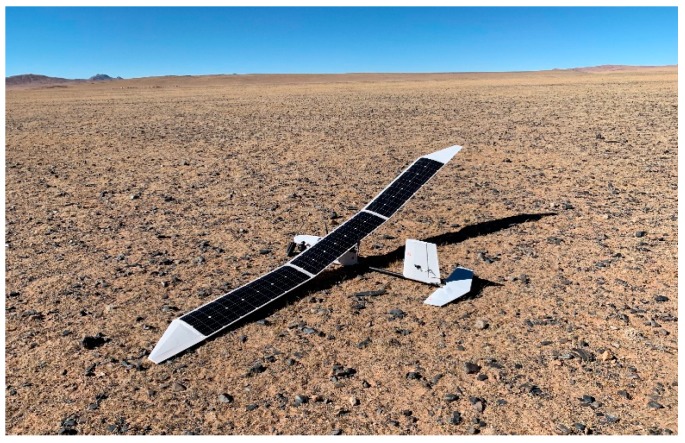
Small hand-launched solar-powered UAV (unmanned aerial vehicle).

**Figure 2 sensors-19-04627-f002:**
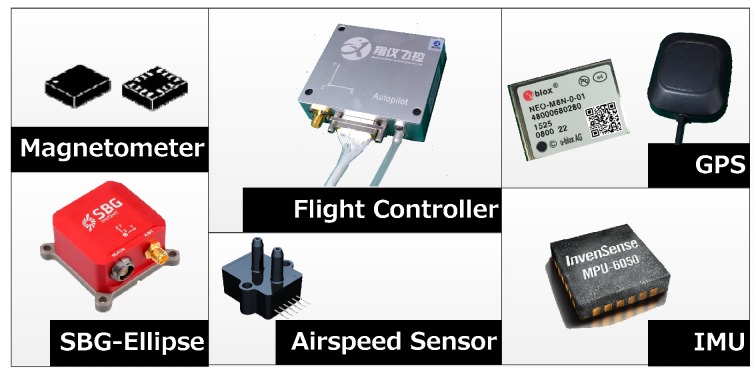
Low-cost flight controller, GPS and (inertial measurement unit) IMU..

**Figure 3 sensors-19-04627-f003:**
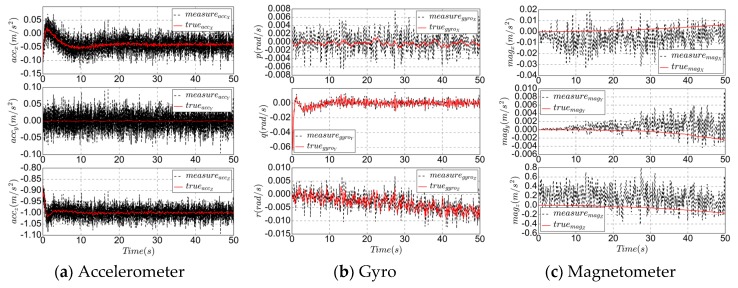
True states and measurement states of low-cost sensors.

**Figure 4 sensors-19-04627-f004:**
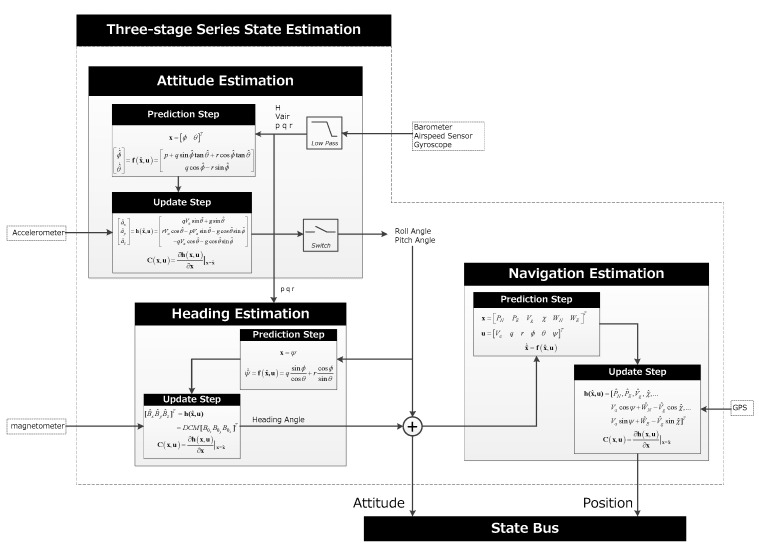
Three-stage series state estimation algorithm structure.

**Figure 5 sensors-19-04627-f005:**
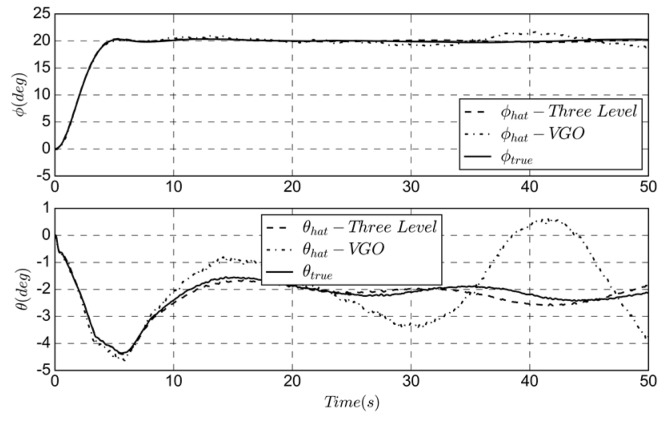
Comparison of true and estimated roll and pitch angle.

**Figure 6 sensors-19-04627-f006:**
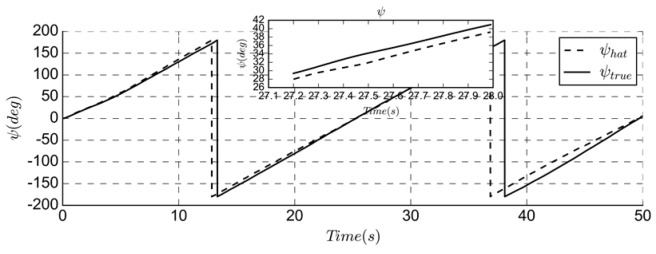
Comparison of true and estimated yaw angle.

**Figure 7 sensors-19-04627-f007:**
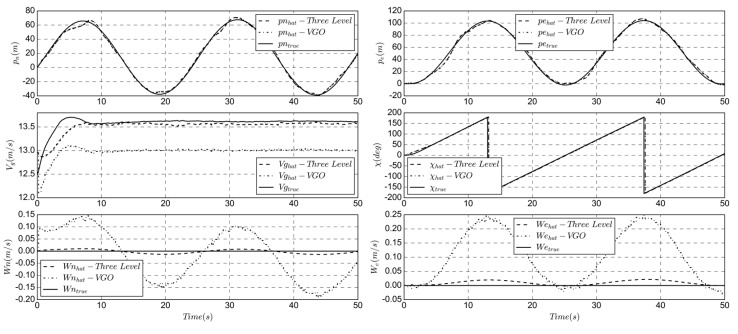
Comparison of true and estimated location, groundspeed and course.

**Figure 8 sensors-19-04627-f008:**
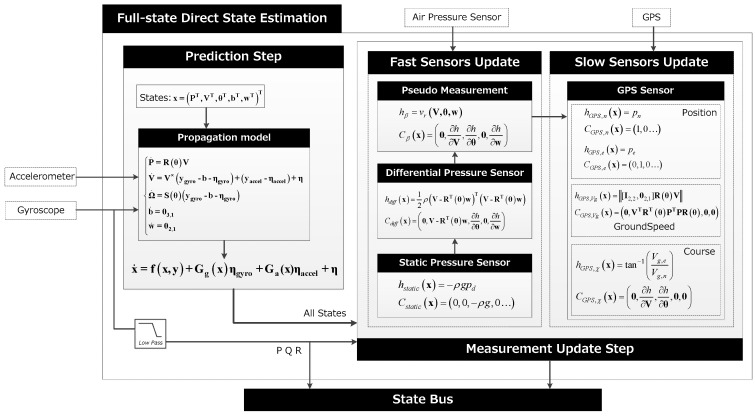
Full-state direct state estimation structure.

**Figure 9 sensors-19-04627-f009:**
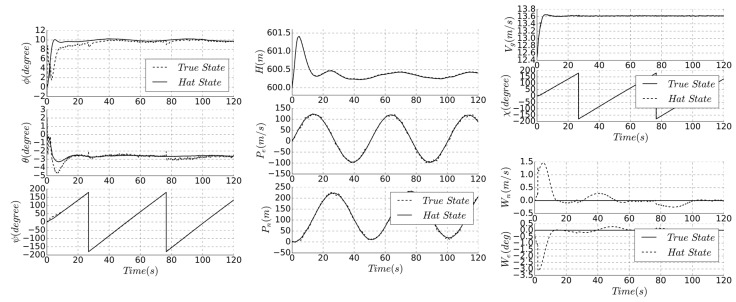
Comparison of true and estimated state.

**Figure 10 sensors-19-04627-f010:**
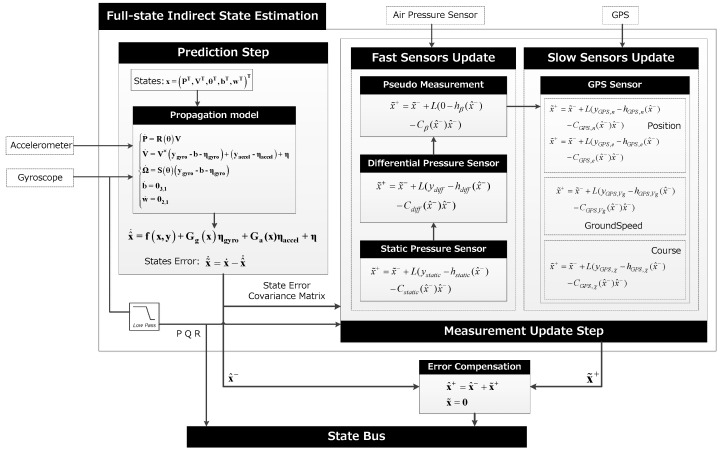
Full-state indirect state estimation structure.

**Figure 11 sensors-19-04627-f011:**
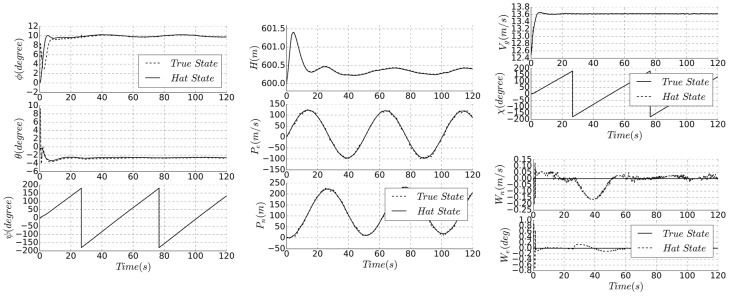
Comparison of true and estimated state.

**Figure 12 sensors-19-04627-f012:**
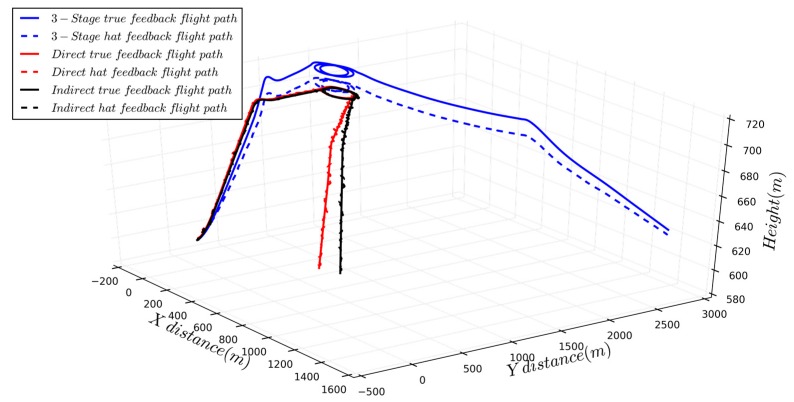
Flight trajectory comparison of different state estimation algorithms.

**Figure 13 sensors-19-04627-f013:**
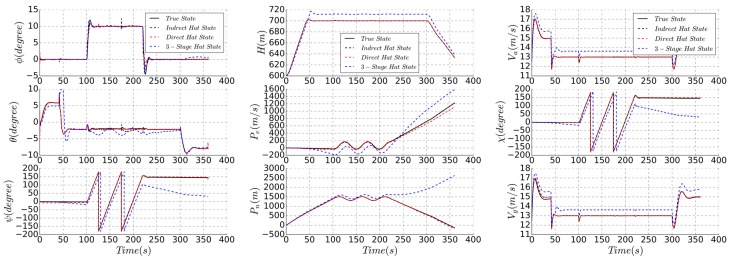
Different mission phases simulation.

**Figure 14 sensors-19-04627-f014:**
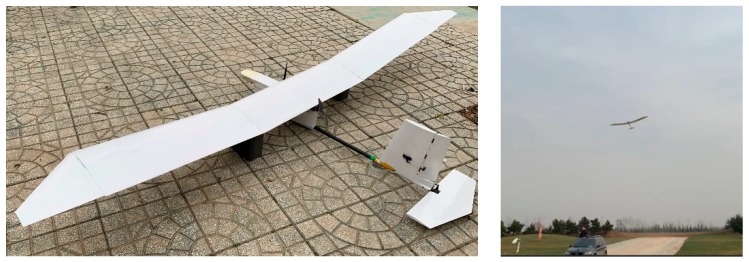
Field flight test of a small hand-launched UAV.

**Figure 15 sensors-19-04627-f015:**
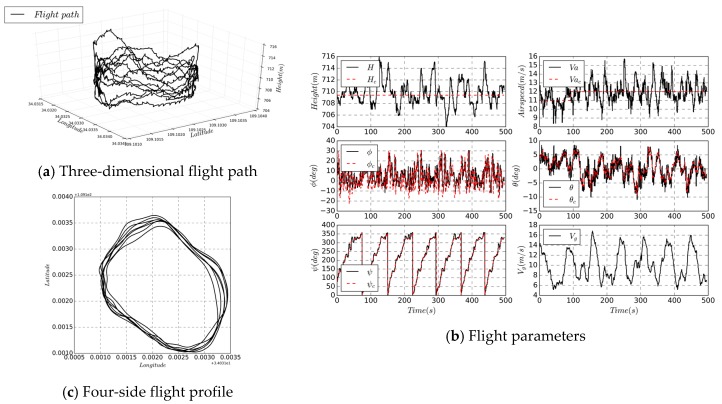
Flight path and state parameters during cruise process.

**Figure 16 sensors-19-04627-f016:**
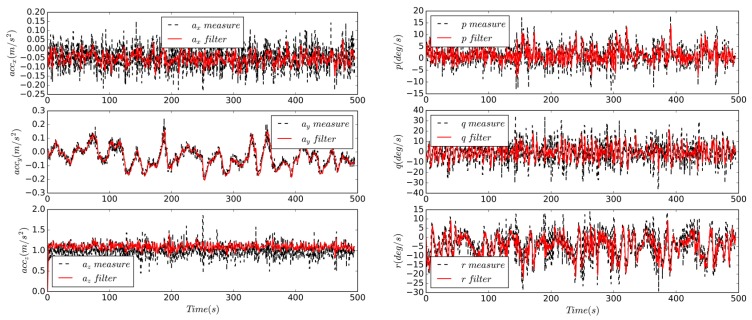
Comparison of IMU measurement and filtered data.

**Table 1 sensors-19-04627-t001:** Various sensors with measurement performance.

Accel	MPU6050	Ellipse	Gyro	MPU6050	Ellipse	Magneto	LSM303D	Barometer	MS5611	MPXV7002	GPS	NEO-M8N
Dynamic range g	±16	±8	Dynamic range °/sec	±2000	±450	Dynamic range gauss	±12	Dynamic range mbar kPa	10–1200	±2	Horizontal accuracy m	±2.5
Nonlinearity %	0.5	0.2	Nonlinearity%	0.2	0.01	Nonlinearity %	0.5	Accuracy mbar %Vfss	±1.5	±2.5	Update rate Hz	10
Cross-Axis Sensitivity %	±2	0.05	Cross-Axis Sensitivity%	±2	0.05	Cross-Axis Sensitivity%/gauss	1	Error mbar V/kPa	±2.5	±1	Start Time s	26
Bias°/sec-rms	±50/±50/±80	-	Mean variance°/sec-rms	0.05	0.135	Sensitivity %/°	±0.05	Delay Time ms	0.5	1	Sensitivity dbm	−148

**Table 2 sensors-19-04627-t002:** Comparison of the estimate accuracy of three algorithms.

	Roll Angle (deg)	Pitch Angle (deg)	Yaw Angle (deg)	Position (m)	Height (m)	Airspeed (m/s)	Ground Speed (m/s)	GPS Course (deg)	Wind (m/s)
Three-stage series	2.52	3.13	7.19	6.04	5.3	2.4	2.9	10.45	0.3
Full-state direct	1.24	0.4	1.58	2.93	0.98	0.8	0.68	1.62	0.3
Full-state indirect	0.48	0.3	0.61	2.9	0.98	0.5	0.37	1.48	0.15

**Table 3 sensors-19-04627-t003:** Comparison of different state estimation algorithms.

	Algorithm Complexity	Hardware Requirement	Estimated Parameters	Estimation Accuracy	Application
Three-stage series	★★	★	14	★★	Low-cost, small UAV, lower mission requirements.
Full-state direct	★★☆	★☆	19	★★★	Low-cost UAV with high trajectory tracking accuracy.
Full-state indirect	★★★	★★	19	★★★★	Mission trajectory requires high precision and large-scale UAV.

**Table 4 sensors-19-04627-t004:** The statistics of IMU measurement noise and bias in the cruising process.

	Accelerometer (m/s^2^)			Gyroscope (deg/s)		
	x Direction	y Direction	z Direction	p	q	r
Noise	0.0818	0.0193	0.2514	4.1424	5.915	5.146
Bias	0.00424	0.00315	−0.0875	−0.1074	0.0383	0.3928

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
