# Peer review of "Low-Cost Sensors State Estimation Algorithm for a Small Hand-Launched Solar-Powered UAV"

_sensors, 2019, doi:10.3390/s19214627_

Round 1

Reviewer 1 Report

The aim of the work isn't clearly written. If the aim is to demonstrate the possibility of low-cost and light sensors to provide the reliable estimation of the navigation parameters it's done in some sense, however the success must be estimation in relation to the real mission of the UAV. Unfortunately, there is nothing about this matter in the article. It looks like the new toy, but no more. I think authors should pay attention to this matter. Another question is the flight duration, should the control system be able to control the solar charging, I guess yes! But authors don't pay attention to this question. I consider this paper just as intermediate report of some serious research, but not as completed research work.

Reviewer 2 Report

This paper outlines mathematical model of solar powered UAVS for flight controllers and its state using EFK algorithms and constrains size/mass based on accuracy, computation by flight controller. The goal is to Improve accuracy of low-cost sensors using state algorithms. Please describe novelty of the work.

Abstract can be re-written to describe major results of the study, rather than the procedure on experimental/modelling scheme

Introduction:

Line 58- 61 “By maximizing using of….ability are satisfied”, this statement is confusing, please re-write and please provide appropriate reference.

Line 64-65- Please clarify statement – “The nonlinear and …. industrial product”.

Fig. 1 Figure is hard to interpret. The axis titles, captions are not clear. Also, specify which equations of the UAV state are modelled in the figure. How does model uncertainty relates to sensors precision and accuracy? There are 16 Fig. panes without any discussion on relevance and conclusion from figure. Please indicate equations used in model. Are there any meteorological conditions (e.g. temperature gradients, wind speed/humidity gradients) taking into models? In few figure panels- the measured states seem unstable and unbounded.

Equation (1) – certain symbols overwritten (same for all equations in the manuscript).

Line 192- Algorithm block diagram is in Fig. 2 (not 4)

Line 206- Symbols are not clear- overwritten

Line 197- 209: In the linearization model, Eqn. 11-17, is the noise from accelerometer and gyroscope included to estimate the altitude and heading states output? What is the metric being used to compare validity and applicability of results in solar powered UAV applications?

Section 3 (General Comment): What are the computation times for this method- is the response time within requirements?

General comment: The placement of Figs. and their description in text is confusing, for example, on Line 260- simulation results are shown in Fig. 7., where Fig. 5 is placed under this text. This is same situation in the entire manuscript- making it harder to read.

Line 366: Reference 22 should be placed in [ ].

Line 357- 362:  From Fig. 11, accuracy is better for direct method- Please clarify and justify the statement- for low time scales the direct method simulations are within simulation errors. Please clarify the range of acceptable errors and required accuracy.

Section 5: Please outline filed and metrological conditions during the test flight and whether model predicts and measured stated were performed under similar conditions and response time for computations under each state. How the overall optimization of state variable achieved for required trajectory and precision?

General Comment: In the manuscript, better (and more rigorous) description of figures and various panels need to be included.

Round 2

Reviewer 2 Report

Revised manuscript accepted.